# Losing a Close Friend or Family Member Due to COVID-19 and Mental Health among LGBTQ Youth

**DOI:** 10.3390/ijerph20126129

**Published:** 2023-06-15

**Authors:** Jonah P. DeChants, Myeshia N. Price, Ronita Nath, Carrie K. Davis

**Affiliations:** The Trevor Project, P.O. Box 69232, West Hollywood, CA 90069, USA

**Keywords:** LGBTQ youth mental health, COVID-19, COVID-19 loss, suicide

## Abstract

This study examines the association between having lost a close friend or family member to COVID-19 and mental health outcomes among lesbian, gay, bisexual, transgender, or queer (LGBTQ) youth. Data come from 33,993 respondents in the United States, ages 13 to 24, who completed an online survey of LGBTQ youth mental health. Multivariate logistic regression was used to determine the adjusted odds of recent anxiety, depression, considering, or attempting suicide in the past year based on whether or not the youth reported having lost a close friend or family member to COVID-19. Among the full sample, experiences of COVID-19 loss were associated with recent anxiety (adjusted odds ratio (aOR) = 1.29, 95% confidence interval (CI) = 1.20–1.40), recent depression (aOR = 1.23, 95% CI [1.15, 1.32]), seriously considering suicide in the past year (aOR = 1.22, 95% CI (1.14, 1.30)), and attempting suicide in the past year (aOR = 1.55, 95% CI (1.41, 1.69)). These findings highlight the urgent need for investment in low-barrier, affirming mental health services for LGBTQ youth who have experienced COVID-19 loss to support their grief, overall mental health, and healthy development.

## 1. Introduction

While numbers are ever-changing as the disease continues to infect, disable, and kill thousands of Americans, over 1,100,000 lives have been lost as a result of the coronavirus disease of 2019 (COVID-19) pandemic in the United States (U.S.) to date [1]. When including excess deaths, it is estimated that each COVID-19 death leaves behind an average of nine surviving individuals, resulting in as many as 9,900,000 Americans impacted by COVID-19 loss and bereavement [2]. As of February 2021, an estimated 37,000 to 43,000 children aged 0 to 17 have lost at least one parent due to COVID-19, with nearly three-quarters of them being adolescents between the ages of 10 and 17 [3]. Research has documented the pandemic’s disproportionate impact on various demographic groups and communities [4,5], and has also begun to explore the pandemic’s impact on several domains of youth development, including academic achievement [6], social isolation [7,8,9] and mental well-being [10,11,12]. However, there has been limited empirical research focused on the impact of losing a close friend or family member to COVID-19 on youth, and there are currently no estimates on the rates of COVID-19 loss and its impact specifically among LGBTQ youth.

The existing research on the impact of COVID-19 loss among adults finds that not only does COVID-19 loss amplify psychological distress [13], but it also may be different when compared to other forms of loss. Findings show that those who are bereaved from COVID-19 report higher levels of grief than people who experienced a loss due to natural causes [14]. Due to potential concerns around transmission, COVID-19 suspended many traditional grief rituals (e.g., families were unable to spend time with the deceased’s body, or had to cremate rather than bury their loved one). The inability to appropriately grieve likely contributed to the experience of prolonged grief among those who lost loved ones to COVID-19 [15]. According to Mohammadi et al. [16], people who lost loved ones to COVID-19 have reported experiencing emotional shock, guilt, rumination, bitter farewell, unconventional burials, and concerns about non-religious burial practices. These experiences may be less common in other forms of loss. Furthermore, and particularly given the economic upheaval and stigma attached to becoming infected in the earlier stages of the pandemic, people have expressed fears about financial insecurity, stigmatization, and complications in social interactions after having experienced a COVID-19 loss [16].

Grief of all varieties is complicated by the COVID-19 pandemic. Families who experienced loss during the pandemic reported a number of challenges–being separated from the relative prior to their death, isolation from peers and other family members, and disruption to daily routines and wider support networks–and complications in family support and coping as both parents and children experienced grief together [9]. Youth who experience bereavement have an increased risk for adverse outcomes, such as anxiety, depression, externalizing behaviors, conduct problems, and substance use [8,9]. Taken together, these findings suggest that COVID-19 loss may have a uniquely harmful impact on youth well-being, both currently and across the lifespan.

These adverse effects of bereavement may have an additional impact on LGBTQ youth, who already report poorer mental health than their straight, cisgender peers [17,18,19,20]. LGBTQ youths’ disproportionately high rates of mental health symptoms are frequently the result of the increased stress, discrimination, and violence that they encounter in a society governed by anti-LGBTQ norms and policies [21,22,23]. Further, initial research suggests that LGBTQ youth have been impacted by unique stressors during the COVID-19 pandemic. For LGBTQ youth living in homes where it is not safe to be out as LGBTQ, the loss of school and peer networks during school closures are particularly stressful [24]. LGBTQ college students who reported that their family did not support their LGBTQ identity were more likely to report frequent psychological distress during the pandemic compared to their LGBTQ peers whose parents did support their LGBTQ identity [25]. 

Based on the available literature, it is likely that COVID-19 loss, and the unique mental health impact connected to it, are intensified for LGBTQ youth, particularly those with multiple marginalized identities [26] who are already confronted with daily stressors due to exposure to anti-LGBTQ discrimination, rhetoric, and harassment. The current study aims to explore rates of having lost a close family member or friend to COVID-19, and the associated mental health outcomes of such a loss for LGBTQ youth, using a national, cross-sectional sample of 33,993 LGBTQ youth. 

## 2. Materials and Methods

### 2.1. Procedure

A non-probability sample of 33,993 LGBTQ youth ages 13 to 24 residing in the United States was collected using an online survey from September 2021 to December 2021. Potential respondents were recruited via targeted ads on social media (Facebook, Instagram, and SnapChat). Participants were defined as being LGBTQ if they identified with a sexual orientation other than heterosexual, a gender identity other than cisgender, or both. Targeted recruitment with respect to race/ethnicity, gender identity, and geography was conducted to ensure adequate sample representation. Qualified respondents completed an online questionnaire with a maximum of 142 questions. Research materials were reviewed and approved by an independent Institutional Review Board, Solutions IRB. Survey participation was voluntary and informed consent was obtained. A waiver of parental consent for youth aged 13 to 17 years was obtained, as the research posed a minimal risk, and contacting parents could have created harm for youth who were not out to their parents about their LGBTQ identity. Demographic characteristics of the sample can be found in Table 1. 

### 2.2. Measures

#### 2.2.1. COVID-19 Loss

Youths’ experience of loss due to COVID-19 was assessed via a survey item which asked, “Has a close family member or friend died due to COVID-19?” Response options included (1) No, and (2) Yes.

#### 2.2.2. Mental Health Symptoms and Suicide Risk

Recent symptoms of anxiety were assessed using an adapted version of the Generalized Anxiety Disorder Assessment (GAD-7) [27]. The survey used four items from the GAD-7, asking respondents to report how often they have been bothered by: “Feeling nervous, anxious, or on edge”, “Not being able to stop or control worrying”, “Feeling afraid as if something awful might happen”, “Trouble relaxing”. Response options for all items are (1) Not at all, (2) Several days, (3) More than half the days, and (4) Nearly every day. Scores were then re-coded and dichotomized with a mean score of three or more being indicative of recent symptoms of anxiety.

Recent symptoms of depression were assessed using an adapted version of the Patient Health Questionnaire (PHQ-9) [28]. The survey used four items from the GAD-7, asking respondents to report how often they have been bothered by: “Little interest or pleasure in doing things”, “Feeling down, depressed, or hopeless”, “Feeling sad”, and “Feeling guilty when bad things happen”. Response options for all items are (1) Not at all, (2) Several days, (3) More than half the days, and (4) Nearly every day. Scores were then re-coded and dichotomized based on a mean score of three or more being indicative of recent symptoms of depression.

Items assessing suicide risk in the last year were from the Centers for Disease Control and Prevention’s (CDC) Youth Risk Behavior Survey (YRBS) [29]. Seriously considering suicide in the past year was assessed using a survey item reading, “During the past 12 months, did you ever seriously consider attempting suicide?” Response options included (1) No, and (2) Yes. Respondents who answered yes were then asked, “During the past 12 months, how many times did you actually attempt suicide?” Response options included (1) 0 times, (2) 1 time, (3) 2 or 3 times, (4) 4 or 5 times, or (5) 6 or more times. Respondents who reported none–as well as those who reported that they had not seriously considered suicide–were coded as 0, and respondents who reported one or more were coded as 1.

#### 2.2.3. Access to Mental Health Services

Access to mental health services in the past 12 months was assessed via a survey item that asked, “In the past 12 months, have you wanted psychological or emotional counseling from a counselor or mental health professional?” Response options included (1) No, (2) Yes, but I didn’t get it, and (3) Yes, and I got it. Respondents who reported not desiring mental health services or desiring them but not receiving them in the past 12 months were coded as 0, and respondents who reported that they had received mental health services in the past 12 months were coded as 1.

#### 2.2.4. Sociodemographic Covariates

Sociodemographic variables included age, gender identity (cisgender girl/woman, cisgender boy/man, transgender girl/woman, transgender boy/man, nonbinary assigned male at birth, nonbinary assigned female at birth, gender questioning assigned male at birth, gender questioning assigned female at birth), sexual orientation (gay/lesbian, bisexual, pansexual, queer, questioning, or heterosexual), race/ethnicity (Native/Indigenous, Asian American/Pacific Islander, Black/African American, Latinx, Multiracial, Middle Eastern or North African, or White), socioeconomic status (just able to meet basic needs or less, more than able to meet basic needs), and U.S. Census Region (Northeast, South, Midwest, West).

### 2.3. Analyses

SPSS 28 was used to conduct all analyses [30]. Chi-square tests of independence were used to examine differences in rates of COVID-19 loss by demographic characteristics. Multivariate logistic regression was used to separately determine the relative odds of recent anxiety, recent depression, considering suicide in the past 12 months, and of attempting suicide in the past 12 months among LGBTQ youth who had lost a close friend or family member to COVID-19 compared to those who had not, adjusting for age, gender identity, sexual identity, race/ethnicity, socioeconomic status, and U.S. region.

## 3. Results

### 3.1. Bivariate Analysis

Rates of COVID-19 loss can be found in Table 2. Overall, 5615 youth, or 18% of the sample, reported losing a friend or close family member due to COVID-19. LGBTQ youth ages 13 to 17 reported significantly higher rates of COVID-19 loss (18.8%) compared to LGBTQ youth ages 18 to 24 (16.0%). Across gender, nonbinary youth who were assigned female at birth reported the highest rates of COVID-19 loss (19.5%) compared to youth of other genders. Pansexual youth reported the highest rates of COVID-19 loss (20.7%) compared to youth of other sexual orientations. Native and Indigenous youth reported the highest rates of COVID-19 loss (27.1%), followed by Latinx youth (25.2%), Middle Eastern and North African youth (24.3%), Black and African American youth (22.2%), Multiracial youth, (19.1%), Asian American and Pacific Islander youth (15.5%), and White youth (14.1%). Youth who reported just being able to meet their basic economic needs reported significantly more COVID-19 loss (22.9%) than youth who said they had more than enough to meet their basic needs (16.3%). Youth in the South reported the highest rates of COVID-19 loss (20.7%) compared to youth in other Census Regions.

### 3.2. Multivariate Logistic Regression Analysis

Adjusted logistic regression was used to examine the association between COVID-19 loss and mental health symptoms, controlling for age, gender identity, sexual identity, race/ethnicity, income, and Census Region. Findings of these logistic regressions can be found in Table 3. Among the sample, LGBTQ youth who reported COVID-19 loss had significantly increased odds of reporting recent anxiety symptoms (aOR = 1.29, 95% CI [1.20, 1.40]), recent major depression symptoms (aOR = 1.23, 95% CI [1.15, 1.315]), considering suicide in the past year (aOR = 1.22, 95% CI [1.14, 1.30]), and attempting suicide in the last year (aOR = 1.55, 95% CI [1.41, 1.69]) compared to LGBTQ youth who had not experienced a COVID-19 loss. 

The sample was then split by whether or not youth had received mental health services in the last twelve months to examine if receipt of mental health services had an impact on the associations between COVID-19-related loss and mental health outcomes. LGBTQ youth who had experienced a COVID-19 loss and had not received any mental health services in the past 12 months reported significantly higher odds of reporting recent anxiety (aOR = 1.33, 95% CI [1.21, 1.45]), recent depression (aOR = 1.22, 95% CI [1.13, 1.33]), considering suicide in the past year (aOR = 1.28, 95% CI [1.18, 1.39]), and attempting suicide in the last year (aOR = 1.77, 95% CI [1.58, 1.98]), compared to LGBTQ youth who had neither experienced a COVID-19 loss nor received mental health services in the past 12 months. LGBTQ youth who had experienced a COVID-19 loss and had received any mental health services in the past 12 months reported significantly higher odds of reporting recent anxiety (aOR = 1.21, 95% CI [1.05, 1.40]), recent depression (aOR = 1.24, 95% CI [1.09, 1.39]), and attempting suicide in the last year (aOR = 1.29, 95% CI [1.11, 1.50]), compared to LGBTQ youth who had not experienced a COVID-19 loss and had received mental health services in the past 12 months.

## 4. Discussion

These findings highlight the urgent need for accessible, affirming COVID-19 loss and bereavement support for LGBTQ youth. Almost one in five (18%) LGBTQ youth in our sample reported having lost a close friend or family member to COVID-19. These data were collected about a year and a half into the pandemic, and given the disease’s continued infection and mortality rates, it is likely that the rate of COVID-19 loss will go up over time. 

Our bivariate findings point to important demographic disparities in COVID-19 loss among LGBTQ youth. LGBTQ youth ages 13 to 17 reported higher rates of COVID-19 loss than their older LGBTQ peers ages 18 to 24. This highlights the need for bereavement support that is specifically designed with youths’ developmental stages in mind. LGBTQ youth who reported just being able to meet their basic economic needs also reported higher rates of COVID-19 loss than their more economically secure peers. As many LGBTQ youth experience economic barriers in accessing mental health care, [31] there is an important need for low-cost and low-barrier mental health care to help LGBTQ young people who have experienced COVID-19 loss to support their grief. An example of a policy that has attempted to increase youth access to mental health care in light of the pandemic is the state of Colorado’s “I Matter” program, which provides six free therapy sessions to all youth residents under the age of 18. [32]. Evaluation research is needed to determine the impact of this program on youth access; however, it is a promising start. 

LGBTQ youth of color reported higher rates of having lost a close friend or family member to COVID-19 compared to their White LGBTQ peers. These findings align with previous scholarship which has documented the disproportionate rates of COVID-19 infection and mortality among communities of color in the U.S. [4,5]. Our findings emphasize the need for COVID-19 loss supports that are culturally-competent. Rituals of grief can vary widely among racial and ethnic communities; therefore, LGBTQ youth of color who have experienced COVID-19 loss would benefit from grief support which is inclusive of and grounded in all of their identities–LGBTQ, racial, cultural, and otherwise. 

Finally, logistic regression findings show that having lost a close friend or family member to COVID-19 is associated with increased odds of recent symptoms of anxiety, recent symptoms of depression, seriously considering suicide in the past year, and attempting suicide in the past year among the full sample of LGBTQ youth. These findings align with previous scholarship documenting the negative impacts of both COVID-19 loss [13,14,16] and general bereavement [8,9] on mental health. Given LGBTQ youths’ already disproportionately high rates of mental health symptoms and suicide risk [17,18,19,20], it is possible that LGBTQ youth who have experienced COVID-19 loss are situated at an intersection of risk which can pose grave threats to their mental health and safety. 

LGBTQ youth who had not received any mental health services in the past 12 months reported higher odds of their mental health being negatively impacted by a COVID-19 loss. This finding is especially apparent in regards to attempting suicide, where they reported 77% higher odds of attempting suicide in the past year compared to their peers who had both not received mental health services and not experienced a COVID-19 loss. Odds of negative mental health outcomes due to COVID-19 loss were generally lower or non-significant for LGBTQ youth who had received mental health services in the past 12 months, with the notable exception of depression. These findings indicate that receipt of mental health services may mitigate some of the negative effects of COVID-19 loss among LGBTQ youth, further highlighting the urgent need for accessible and affirming mental health support for LGBTQ youth who have experienced COVID-19 loss. Existing services that seek to support youth experiencing COVID-19 loss must be accepting and affirming of LGBTQ young people, so that they can feel safe discussing all aspects of their lives and identities with mental health professionals [17].

### Limitations

This study boasts a large, diverse sample of LGBTQ youth from across the U.S.; however, some limitations should be noted. The study’s cross-sectional design precludes any determination of causation or directionality of the findings. Future research should employ longitudinal methods to examine the long-term impact of COVID-19 loss over the lifespan. Our survey also relies on self-reported measures, which in the case of COVID-19 loss, may be obscured by youths’ lack of information about their loved ones’ deaths. Youths’ families may have tried to protect them by not sharing details about their loved ones’ deaths, or the co-occurrence of COVID-19 and other illnesses or conditions may have caused families to not know for certain if their loved one died due to COVID-19. We also did not collect information about the nature of the relationships between youth and their “close friends or family members” lost to COVID-19. Therefore, we cannot evaluate how the loss of different types of close relationships (e.g., loss of a parent vs. loss of a friend) impacts mental well-being. Future research should consider potential differences in these relationships and their impact on mental health, for example, examining the difference between losing a custodial caregiver or a more distant relative. Future research must also continue to include LGBTQ youth from under-studied backgrounds, such as LGBTQ youth in rural areas, LGBTQ youth of color, and LGBTQ youth with disabilities in examinations of the impact of COVID-19 loss. 

## 5. Conclusions

This study is the first to examine the impact of COVID-19 loss on the mental health and suicide risk of LGBTQ youth. These findings highlight the urgent need for accessible, LGBTQ-inclusive, culturally-competent mental health services and grief support for LGBTQ youth. Adults working with youth–whether in schools, clinical, or community-based settings–must be aware of the real potential for COVID-19 loss among this group, and its potential impact on their mental health and development. While the COVID-19 pandemic has ceased to define daily life for many of us, it is imperative that we support the young people whose lives and families have been forever changed by it. Accessible and competent mental health services can support LGBTQ youth as they grow into happy and healthy adults.

## Figures and Tables

**Table 1 ijerph-20-06129-t001:** Sample Demographics.

	*n* = 33,993 (%)
Age (*n* = 33,993)	
13–17	21,227 (62.4)
18–24	12,766 (37.6)
Gender Identity (*n* = 33,219)	
Cisgender boy/man	5093 (15.3)
Cisgender girl/woman	8155 (24.5)
Transgender boy/man	3079 (9.3)
Transgender girl/woman	1574 (4.7)
Nonbinary assigned male at birth	2249 (6.8)
Nonbinary assigned female at birth	9956 (30)
Questioning assigned male at birth	645 (1.9)
Questioning assigned female at birth	2467 (7.4)
Sexual Identity (*n* = 33,841)	
Gay or lesbian	8611 (25.4)
Straight	274 (0.8)
Bisexual	10,410 (30.8)
Queer	3552 (10.5)
Pansexual	6877 (20.3)
Asexual	2919 (8.6)
I am not sure	1198 (3.5)
Race/Ethnicity (*n* = 32,561)	
African American/Black	2159 (6.6)
Asian American/Pacific Islander	1706 (5.2)
Latinx	5543 (17)
Native/Indigenous	352 (1.1)
Middle Eastern/North African	238 (0.7)
Multiracial	4739 (14.6)
White	17,824 (54.7)
Socioeconomic Status (*n* = 31,160)	
Just meeting basic needs	6259 (20.1)
More than meets basic needs	24,901 (79.9)
U.S. Region (*n* = 33,993)	
Northeast	5329 (15.7)
South	11,740 (34.5)
Midwest	7873 (23.2)
West	9051 (26.6)
Mental Health Access (*n* = 31,137)	
No mental health services in the last 12 months	22,377 (67.1)
Received mental health services in the last 12 months	10,960 (32.9)
Outcome Variables	
Recent anxiety (*n* = 33,800)	24,642 (72.9)
Recent depression (*n* = 33,625)	19,440 (57.8)
Seriously considered suicide (*n* = 31,132)	13,949 (44.8)
Attempted suicide (*n* = 30,694)	4378 (14.3)

**Table 2 ijerph-20-06129-t002:** Rates of COVID-19 loss by demographic characteristics.

	COVID-19 Loss *n* = 31,675 (%)
Age (*n* = 31,675)	
13–17	3673 (18.8)
18–24	1942 (16.0)
Gender Identity (*n* = 31,000)	
Cisgender boy/man	782 (16.3)
Cisgender girl/woman	1371 (18.1)
Transgender boy/man	517 (18.0)
Transgender girl/woman	198 (13.2)
Nonbinary assigned male at birth	333 (15.6)
Nonbinary assigned female at birth	1797 (19.5)
Questioning assigned male at birth	85 (13.9)
Questioning assigned female at birth	412 (18.2)
Sexual Identity (*n* = 31,550)	
Gay or lesbian	1391 (17.2)
Straight	39 (15.2)
Bisexual	1684 (17.4)
Queer	566 (16.9)
Pansexual	1315 (20.7)
Asexual	397 (14.6)
I am not sure	194 (17.8)
Race/Ethnicity (*n* = 30,407)	
African American/Black	439 (22.2)
Asian American/Pacific Islander	244 (15.5)
Latinx	1286 (25.2)
Native/Indigenous	89 (27.1)
Middle Eastern/North African	53 (24.3)
Multiracial	850 (19.1)
White	2366 (14.1)
Socioeconomic Status (*n* = 29,236)	
Just meeting basic needs	1330 (22.9)
More than meets basic needs	3811 (16.3)
U.S. Region (*n* = 31,675)	
Northeast	750 (15.1)
South	2253 (20.7)
Midwest	1139 (15.5)
West	1473 (17.4)
Mental Health Access (*n* = 31,137)	
No mental health services in the last 12 months	3859 (18.6)
Received mental health services in the last 12 months	1672 (16.2)
Outcome Variables	
Recent anxiety (*n* = 31,530)	9826 (78.3)
Recent depression (*n* = 31,390)	8100 (64.7)
Seriously considered suicide (*n* = 29,192)	6251 (53.3)
Attempted suicide (*n* = 28,809)	2134 (18.5)

**Table 3 ijerph-20-06129-t003:** Bivariate logistic regression of COVID-19 loss on mental health.

		aOR (95% CI)	*p*-Value
Full sample			
	Recent anxiety	1.29 (1.20–1.40)	<0.001
	Recent depression	1.23 (1.15–1.32)	<0.001
	Seriously considered suicide	1.22 (1.14–1.30)	<0.001
	Attempted suicide	1.55 (1.41–1.69)	<0.001
No mental health services in the last 12 months			
	Recent anxiety	1.33 (1.21–1.45)	<0.001
	Recent depression	1.22 (1.13–1.33)	<0.001
	Seriously considered suicide	1.28 (1.18–1.39)	<0.001
	Attempted suicide	1.77 (1.58–1.98)	<0.001
Received mental health services in the last 12 months			
	Recent anxiety	1.21 (1.05–1.40)	<0.01
	Recent depression	1.24 (1.09–1.39)	<0.001
	Seriously considered suicide	-	-
	Attempted suicide	1.29 (1.11–1.50)	<0.001

Adjusted for age, gender identity, sexual identity, race/ethnicity, socioeconomic status, and Census Region.

## Data Availability

Data sharing is not available due to privacy and ethical restrictions.

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
