# Peer review of "Losing a Close Friend or Family Member Due to COVID-19 and Mental Health among LGBTQ Youth"

_ijerph, 2023, doi:10.3390/ijerph20126129_

Round 1
Reviewer 1 Report
This manuscript focused on the impact of COVID-19 loss on the mental health and suicide risk in a large and diverse sample of LGBTQ youth is very interesting. Although the authors mention some limitations, I believe that there are other points that should be mentioned.
1. How do you ensure that the responses of the younger population (13 to 17 years) on anxiety and depression are valid? How to ensure that in this age group they are clear about the symptomatology that covers each of these health conditions?
2. Did you apply an anxiety and depression scale?
3. Did the population group studied have a previous diagnosis of these mental health conditions by a health professional?
4. How to differentiate in the full sample that mental health disorders appeared after the loss of a friend or family member due to covid 19?
Author Response
Reviewer #1
This manuscript focused on the impact of COVID-19 loss on the mental health and suicide risk in a large and diverse sample of LGBTQ youth is very interesting. Although the authors mention some limitations, I believe that there are other points that should be mentioned.
Response:
Thank you very much for your thoughtful review of the manuscript. We appreciate your feedback and have addressed your comments below.
- How do you ensure that the responses of the younger population (13 to 17 years) on anxiety and depression are valid? How to ensure that in this age group they are clear about the symptomatology that covers each of these health conditions?
Response:
Our measure of Anxiety and Depression did not rely on youths’ self-report, but on scales which have been validated and widely used among youth and adolescents in the United States. We have added descriptions of these measures to section 2.2. Measures in the manuscript.
Anxiety was assessed using an adapted version of the Generalized Anxiety Disorder Assessment (GAD-7) and asks respondents to report how often they have been bothered by: “Feeling nervous, anxious, or on edge”, “Not being able to stop or control worrying”, “Feeling afraid as if something awful might happen”, “Trouble relaxing”. Response options for all items are 1) Not at all, 2) Several days, 3) More than half the days, and 4) Nearly every day. Scores were then dichotomized for a mean score of three or more being indicative of a clinically relevant anxiety disorder.
Depression was assessed using an adapted version of the Patient Health Questionnaire (PHQ-9). This scale also asks respondents to report how often they have been bothered by: “Little interest or pleasure in doing things”, “Feeling down, depressed, or hopeless”, “Feeling sad”, “Feeling guilty when bad things happen”. Like the GAD-2, response options for all items are 1) Not at all, 2) Several days, 3) More than half the days, and 4) Nearly every day. Scores were then re-coded and dichotomized based with a mean score of three or more being indicative of recent symptoms of depression.
- Did you apply an anxiety and depression scale?
Response:
Yes, as described in our previous response we used adapted versions of the GAD-7 to assess recent anxiety and the PHQ-9 to assess recent depression.
- Did the population group studied have a previous diagnosis of these mental health conditions by a health professional?
Response:
We did not assess previous diagnoses of mental health conditions by a health professional in this survey, therefore we do now know who among the sample may have received a diagnosis. We chose to focus on assessments of current symptoms since many younger youth in the sample may not know if they have received a mental health diagnosis if it happened earlier in their childhood.
- How to differentiate in the full sample that mental health disorders appeared after the loss of a friend or family member due to covid 19?
Response:
Due to both the lack of data on mental health diagnoses and the cross-sectional nature of this survey, we are not able to differentiate in the full sample between mental health disorders that appeared after the loss of a friend or family member due to COVID-19. In our findings we are not claiming any chronological or causal relationship between COVID loss and mental health symptoms, we are simply reporting the association observed in our point-in-time data between COVID-loss and increased likelihood of mental health symptoms and suicide risk.
Reviewer 2 Report
Dear Authors
The paper is characterized by originality, and it is well structured. COVID-19 pandemic has affected the daily life of many patients with different backgrounds. Consequently, it seems perfectly reasonable that it has affected the quality of life of LGBTQ youth
I have some minor observations:
-On line 104: Prevention’s (CDC) Youth Risk Behavior Survey (YRBS). (27) Seriously considering.
I think you should move the number.
-On line 131: SPSS 28 was used to conduct all analyses.28 Chi-square tests of independence were
I think you should move the number.
-You should change the number of the table (Findings of these logistic regressions can be found in Table 2)
- For a more robust statistical analysis, I suggest using other tests such as a Person correlation of the different variables.
Author Response
Reviewer #2
Dear Authors
The paper is characterized by originality, and it is well structured. COVID-19 pandemic has affected the daily life of many patients with different backgrounds. Consequently, it seems perfectly reasonable that it has affected the quality of life of LGBTQ youth
Response:
Thank you very much, we are glad you also see the value of these findings and appreciate your feedback.
I have some minor observations:
-On line 104: Prevention’s (CDC) Youth Risk Behavior Survey (YRBS). (27) Seriously considering. I think you should move the number.
Response:
Thank you for pointing this out, we have made the footnote a superscript as consistent with the rest of the manuscript.
-On line 131: SPSS 28 was used to conduct all analyses.28 Chi-square tests of independence were. I think you should move the number.
Response:
Thank you for pointing this out, we have made the footnote a superscript as consistent with the rest of the manuscript.
-You should change the number of the table (Findings of these logistic regressions can be found in Table 2)
Response:
- For a more robust statistical analysis, I suggest using other tests such as a Person correlation of the different variables.
Response:
Thank you for your suggestion to use Pearson correlation for a more robust statistical analysis. We kindly request that you clarify your suggestion for using Pearson correlation in the context of our study, as we may not fully understand the intended application. We are confident that our chosen statistical methods are both valid and relevant for our research question. The chi-square test is well-suited for examining differences in categorical variables, such as the rates of COVID-19 loss by demographic characteristics. Furthermore, the multivariate logistic regression allows us to assess the relationship between multiple independent variables and a binary dependent variable (e.g., considering or attempting suicide) while adjusting for potential confounding factors. Pearson correlation, on the other hand, is more applicable to continuous variables and would not be suitable for our categorical data
Reviewer 3 Report
This is an excellent study done on a large sample, and the topic is relevant and important. What is missing is maybe more references to other studies on mental health in the COVID-19 situation - what makes the LGBTs unique? Different, I suppose, yes. But how much different? I think that suicidal thoughts, from my psychological point of view, may be the key. But what about suicidal thoughts in "normal" conditions"? Every loss is a trauma; therefore, linking it with COVID-19 is a little overinterpreted. So you may consider those remarks and other reviewers' opinions when working on the final paper. Congratulations! Very well done study. Maybe also may be the cultural factor included here...the culture of the U.S.... Thinking bout the probable results if done in a more hostile environment for LGBTs in Poland, my country...
Author Response
Reviewer #3
This is an excellent study done on a large sample, and the topic is relevant and important.
Response:
Thank you for your review and feedback.
What is missing is maybe more references to other studies on mental health in the COVID-19 situation - what makes the LGBTs unique? Different, I suppose, yes. But how much different? I think that suicidal thoughts, from my psychological point of view, may be the key. But what about suicidal thoughts in "normal" conditions"? Every loss is a trauma; therefore, linking it with COVID-19 is a little overinterpreted.
Response:
Thank you for this feedback. We discussed LGBGTQ youths’ unique psychological stressors and outcomes in normal conditions and related to the COVID-19 pandemic in the second-to-last paragraph of the Introduction. We noted that even prior to the pandemic LGBTQ youth report elevated levels of mental health symptoms and suicide risk due to minority stress and cited previous work on how the pandemic may have uniquely impacted LGBTQ youth. We agree that every loss is a trauma but this manuscript specifically examines self-reported COVID loss and its association with mental health and suicide risk.
So you may consider those remarks and other reviewers' opinions when working on the final paper. Congratulations! Very well done study. Maybe also may be the cultural factor included here...the culture of the U.S.... Thinking bout the probable results if done in a more hostile environment for LGBTs in Poland, my country…
Response:
Thank you for this feedback, we agree that there are likely cultural factors in play- both in terms of the US’s approach to COVID and in terms of the unique manifestations of anti-LGBTQ bias in each country. While the US is in some ways a global leader in LGBTQ legal rights, we are currently experiencing a massive wave of anti-LGBTQ legislation seeking to limit LGBTQ youths’ rights to healthcare, privacy, and inclusive education. Unfortunately we do not have the expertise on other cultural contexts for us to provide any comparative insights in this manuscript.
Round 2
Reviewer 1 Report
The authors included the anxiety and depression symptom scales. With this information the procedure section was improved.
